# Wavelet Analysis Reveals Phenology Mismatch between Leaf Phenology of Temperate Forest Plants and the Siberian Roe Deer Molting under Global Warming

Heqin Cao [1,2,3,†], Yan Hua [4,†], Xin Liang [2], Zexu Long [2], Jinzhe Qi [2], Dusu Wen [2], Nathan James Roberts [2], Haijun Su [1,3,*] and Guangshun Jiang [2,*]

[1] Forestry College, Guizhou University, Guiyang 550025, China
[2] Feline Research Center of National Forestry and Grassland Administration, College of Wildlife and Natural Protected Area, Northeast Forestry University, Harbin 150040, China
[3] Research Center for Bio-Diversity and Nature Conservation, Guizhou University, Guiyang 550025, China
[4] Guangdong Provincial Key Laboratory of Silviculture, Protection and Utilization, Guangdong Academy of Forestry, Guangzhou 510520, China
* Correspondence: hjsu@gzu.edu.cn (H.S.); jiangguangshun@nefu.edu.cn (G.J.)
† These authors contributed equally to this work.

**Abstract:** Global warming is deeply influencing various ecological processes, especially regarding the phenological synchronization pattern between species, but more cases around the world are needed to reveal it. We report how the forest leaf phenology and ungulate molting respond differently to climate change, and investigate whether it will result in a potential phenology mismatch. Here, we explored how climate change might alter phenological synchronization between forest leaf phenology and Siberian roe deer (*Capreolus pygargus*) molting in northeast China based on a camera-trapping dataset of seven consecutive years, analyzing forest leaf phenology in combination with records of Siberian roe deer molting over the same period by means of wavelet analysis. We found that the start of the growing season of forest leaf phenology was advanced, while the end of the growing season was delayed, so that the length of the growing season was prolonged. Meanwhile, the start and the end of the molting of Siberian roe deer were both advanced in spring, but in autumn, the start of molting was delayed while the end of molting was advanced. The results of wavelet analysis also suggested the time lag of synchronization fluctuated slightly from year to year between forest leaf phenology and Siberian roe deer molting, with a potential phenology mismatch in spring, indicating the effect of global warming on SRD to forest leaf phenology. Overall, our study provides new insight into the synchronization between forest leaf phenology and ungulate molting, and demonstrates feasible approaches to data collection and analysis using camera-trapping data to explore global warming issues.

**Keywords:** leaf phenology; Siberian roe deer; molting; wavelet analysis; phenology mismatch; climate warming

## 1. Introduction

Phenology is the response of the growth, development, and activity rhythm of animals and plants to climatic factors, and is the result of organisms adapting to seasonal changes in the environment over a long period of time [1]. For example, the annual periodic growth of leaf spreading and leaf fall of plants, and the migration, molting, and hibernation of animals are all phenological phenomena. Animals and plants have formed a relatively stable phenological synchronization pattern in the long-term evolution process [2]. Siberian roe deer (SRD) (*Capreolus pygargus*) start molting at the peak of leaf expansion in northern deciduous broadleaf forests [3]. We hypothesized that the synchronization

effect of SRD spring molting and forest leaf phenology (FLP) is the dual needs of nutritional adaptation and concealment during the molting period. Some researchers found that seasonal molting is the response of animals to different seasonal environmental factors [4,5], and the obvious difference in animal coat color between winter and summer is an important reference standard for the study of climate change on animal camouflage mismatch [6,7]. Therefore, animal molting and leaf phenology are ideal features to study the impact of climate change on phenology.

Under the action of global climate change, the changes of interspecific interactions in ecosystems are far more far-reaching than the direct effects of abiotic factors (such as temperature, photoperiod, etc.), and deserve more attention [8]. A large number of studies have shown that the general rhythms of animals and plants responding to climate change is advanced in the spring life history such as leaf spreading, flowering, hatching [9], and bird migration [10], prolonged in the growing season, and have delayed autumn phenology. Climate change will destroy the phenological consistency of intercrop species formed by long-term coevolution to a certain extent and cause an asynchrony between the reproductive phenology of birds and that of insects, thus reducing the reproductive efficiency of birds [11]. Phenological mismatches between snow presence and snowshoe hare *Lepus americanus* pelage coloration could compromise crypsis and lead to elevated predation risk [6]. Although animals and ecosystems have the ability to resist the fluctuation of environmental factors to maintain the stability of the population or ecosystem, they can only play a role in a certain range. If the pace of evolution fails to match that of climate warming [12], the functional efficiency of molt may be undermined.

In recent decades, human activities have accelerated the process of global climate warming. Due to the different temporal and spatial effects and degrees of animal and plants affected by climatic factors, the previously stable synchronization pattern of animal and plant phenology may be broken, which will seriously affect the stability of ecosystem functions [13,14]. However, the research on how plant and animal phenology responds to climate change is rare and limited to traditional research methods. Whether climate change will affect the synchrony of plant and animal phenology and how it will affect it is also unknown. Richardson et al. [15] for the first time used greenness index to monitor spring phenology of temperate deciduous broadleaf forests, and proved that greenness index extracted from digital cameras can accurately reflect canopy conditions, which was then widely used to monitor ecosystems [16]. Nowak et al. [17] quantified and measured the molting of snow sheep (*Oreamnos americanus*) with infrared camera, which was the world's first study on the molting of wild animal populations, making automatic camera monitoring of seasonal molting of wild populations a reality. Therefore, we chose deciduous broadleaf plant and molting characteristics of SRD as research objects, and hypothesized that climate warming will lead to the advance of FLP, while SRD molting time does not change much in line with photoperiod, resulting in a potential phenology mismatch between them. We focused on phenological synchronization based on camera-traps data, and aimed to reflect the adaptability of SRD to the environment, and bring new ideas and suggestions to issues such as global warming combined with the phenology and ecological adaption of animals.

## 2. Materials and Methods

### 2.1. Data Collection

During 2013–2019, we selected a total of 153 camera trap sites (total camera sampling effort of 381,076 days) with a density of one site per 3 km² for animal population surveys. Camera traps were placed near animal trails or mountain ridges, haulage paths, and water sources with a high probability of animal occurrence. We set one infrared camera (LTL-5210A and LTL-6210, Shenzhen Weikexin Science and Technology Development Co., Ltd., Shenzhen, China) per site in the study area (Hunchun Amur tiger National Nature Reserve and Lanjia Nature Reserve, northeast China (Figure 1). To take quality pictures of

roe deer, the preset focus was set at 45–50 cm above the ground and at a horizontal distance of 3–4 m with shooting mode of 3 photos plus 1 video. All camera traps were set for continuous monitoring. Only photos and videos containing SRD were selected for molting measurement. Meanwhile, daily empty photos (meaning not taking any pictures of animals) were selected for leaf phenology analysis.

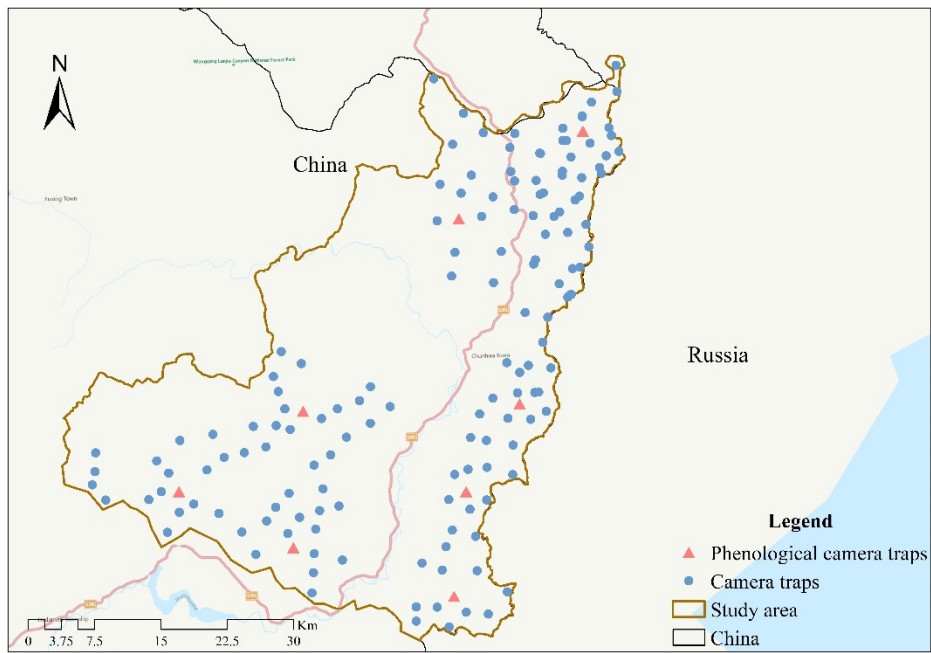

**Figure 1.** Camera trap sites in Hunchun Amur tiger National Nature Reserve and Lanjia Nature Reserve, northeast China.

## 2.2. Forest Leaf Phenology Analysis

We used phenopix package in R [18] to extract green chromatic coordinate (GCC), which is a vegetation index derived from photographic images and quantifies the greenness relative to the total brightness. First, we selected the Regions of Interest (ROI), then digital color numbers were extracted from the ROI of each image and processed to obtain a continuous time series. Then, we extracted GCC. Finally, after filtering the time series, data were fitted with either a double logistic equation or a smoothing curve, on which phenological thresholds (phenophases) were extracted, such as start of growing season (SOS), length of growing season (LOS), end of growing season (EOS) and peak of season position (POP).

## 2.3. Measurement and Determination of Siberian Roe Deer Molting

To estimate the extent of SRD molting, we compared pixel counts of molted versus unmolted areas of SRDs' coats in each photograph using Adobe Photoshop CS6. The specific molting scores can be expressed as follows: molting scores = pixel counts of SRD molting area / total pixel counts of SRD [17]. In spring, winter coat reaching less than 95% is defined as the SRD spring start of molting (SOM). In autumn, winter coat reaching more than 5% is defined as the autumn SOM [9]. Then, we selected the individual molting scores from March 1 to June 30 (spring), and July 1 to October 30 (autumn), using moult package in R [19,20] to model, for estimating duration molting, mean start date molting, and its standard deviation.

### 2.4. Trend Analysis

Taking phenological parameters (start of growing season, end of growing season, start and end date of SRD molting) as research objects, the interannual variation trend of forest leaf phenophase and SRD molting were analyzed. Univariate linear regression was used to analyze the interannual variation of FLP and SRD molting in study area (Formula (1)).

$$y_i = kt_i + b \ (I = 1, 2, 3, \ldots, n) \tag{1}$$

Formula (1) $y_i$ represents the phenological parameters, $t_i$ represents the year corresponding to $y_i$, k represents the regression coefficient, and b represents the regression constant.

k and b were estimated by the least square method. The regression coefficient k represents the interannual variation rate of vegetation phenology, and its absolute value represents the interannual increase/decrease rate of each phenology. When $k > 0$, the phenological parameters were delayed with the increase of years. When $k < 0$, phenological parameters showed an advancing trend with the increase of years.

### 2.5. Relationship between Meteorological Factors and Forest Leaf Phenology

From 2013 to 2019, the meteorological data of the Hunchun Reserve were obtained from the Hunchun Meteorological Station (China Daily Value Dataset of Surface Meteorological Data (V3.0)). The mean temperature (MT), sunshine duration (ssd), and precipitation (pre) in Hunchun were extracted to explore the influence of meteorological factors on FLP (Supplementary Table S1).

We used the optimal period (OP) of Dai [21]: the phenological period is mainly affected by the meteorological factors in the period before the occurrence of the phenological period, and the length of time that a specific meteorological factor may affect a phenological event is defined as the OP (Formula (2)).

$$LP = [BP, EP] \tag{2}$$

LP is the length of the optimal period (days), EP is the end date of LP, represented by the daily ordinal number of year (DOY), and is defined as the annual phenological period (the start and end of the growing season) in this paper. BP is the start date of LP. The value ranges from 1 to EP-1. Then, Pearson correlation coefficient (R) between phenology and meteorological factors in BP-EP was calculated by moving a step of −1 day for each BP from EP-1.

Univariate linear regression analysis and multiple linear regression analysis were carried out on the mean values of meteorological factors in the phenological period and OP to explore the effects and patterns of meteorological factors on phenology.

### 2.6. Relationship between Meteorological Factors and Siberian Roe Deer Molting

We selected the monthly MT, ssd of spring and autumn SOM and end of molting (EOM), MT and ssd on the day of SRD molting, MT and ssd from winter/summer solstice to SOM/ EOM. Then use Spearman correlation analysis to explore the influence of these meteorological factors on SRD molting.

### 2.7. Synchronization Analysis of Forest Leaf Phenology and Siberian Roe Deer Molting

Wavelet analysis is a powerful tool based on Fourier analysis, which can divide a signal (time series) into different oscillatory components with different frequencies (periods). Wavelet analysis makes it possible to track how the different scales associated with the periodic component of the signal change over time. It can not only extract the information of different periodic components in time series, but also reflect the evolution of these periodic components over time [22]. Due to the specific nature of ecological and

environmental time series and the relationship between them, wavelet decomposition has been used many times in ecological research [23–25].

We used the WaveletComp package in R to carry out the continuous wavelet-based analysis of both univariate and bivariate time series [26]. We used the wavelet power spectrum and cross-wavelet power spectrum [23] to define the temporal relationships between FLP and SRD molting. Firstly, the time series of GCC and SRD molting scores were used to carry out univariate wavelet analyses. We reconstructed the time series based on the decomposition results to test the reliability of the analysis and calculated the wavelet power spectrum and mean power curve based on the calculated wavelet coefficients, which indicate the similarity of the target time series and waveforms with different amplitudes and periods. Then, the cross-wavelet analysis was used to explore the temporal synchrony between FLP and SRD molting. We also analyzed the synchrony between paired time series. Phase differences were extracted at a particular time scale when the synchrony of two-time series was significant and persistent. Finally, cross wavelet analysis was carried out on meteorological factors (MT and ssd), FLP and SRD molting to explore their relationships.

## 3. Results

### 3.1. Forest Leaf Phenology Monitor

#### 3.1.1. Key Phenophases of Forest Leaf Phenology

Max, Night and Spline methods were selected to filter data. By comparing the filtering results of the three methods, the results of Max method were selected for subsequent data fitting (Supplementary Figure S1). Then, by analyzing different fitting methods and the extracted vegetation index, an appropriate growth curve fitting extraction method was selected (Supplementary Figures S2 and S3). Finally, the Klosterman-Klosterman method was more suitable for fitting the data and extracting the phenophases in our study. Results showed that the SOS was in late-April and early-May, and the EOS was in mid-October (Table 1).

**Table 1.** Key phenological parameters were extracted by the Klosterman-Klosterman method.

| Method | Year | Start of Growing Season | End of Growing Season | Length of Growing Season | Peak of Season Position |
|--------|------|------------------------|----------------------|-------------------------|------------------------|
| | 2013 | 125 | 281 | 156 | 158 |
| | 2014 | 124 | 282 | 158 | 156 |
| | 2015 | 115 | 280 | 165 | 150 |
| Klosterman | 2016 | 116 | 281 | 166 | 151 |
| | 2017 | 119 | 294 | 174 | 155 |
| | 2018 | 120 | 288 | 168 | 151 |
| | 2019 | 114 | 287 | 172 | 148 |

#### 3.1.2. Interannual Variation Trend of Forest Leaf Phenology 2013–2019

The results show that from 2013 to 2019, the SOS of FLP in our study area was advanced, and the EOS was delayed, so that the LOS was prolonged (Figure 2).

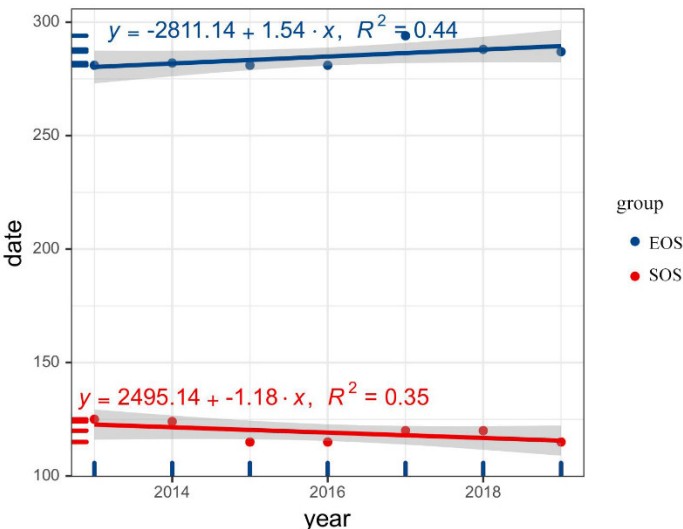

**Figure 2.** Interannual variation trend of forest leaf phenology's start of growing season and end of the growing season in the study area during 2013–2019. EOS: end of growing season, SOS: start of growing season.

### 3.2. Camera-Trap Monitoring of Siberian Roe Deer Molting

From 2013 to 2019, a total of 12,447 SRD photos and 2919 videos, were taken in the study area, with a resolution of 2560 × 1920, 1491 effective photos (flank clearly visible and not affected by light), and 916 effective videos. There were 509 effective molting events (Supplementary Table S2).

### 3.2.1. Start Date and Duration of Siberian Roe Deer Molting

There were 271 and 238 SRD molting events in spring and autumn from 2013 to 2019, respectively (Supplementary Table S3), and differences were observed. The mean SOM of SRD was in mid-April and late-April, and the duration was about 45 days in spring. In autumn, the mean SOM of SRD was in mid-August and at end of August, and the duration was about 40 days (Table 2).

**Table 2.** Siberian roe deers' start of molting and duration (SE = standard error, SD = standard deviation) from 2013–2019.

| Season | Year | Duration ± SE (Day) | Start Date± SE (Day) | SD of Start Date± SE (Day) |
|--------|------|---------------------|----------------------|----------------------------|
| Spring | 2013 | 50.12 ± 4.25 | 108.89 ± 2.56 | 5.11 ± 1.85 |
|        | 2014 | 52.26 ± 1.51 | 116.82 ± 0.65 | 1.78 ± 0.58 |
|        | 2015 | 39.60 ± 2.16 | 119.76 ± 1.64 | 3.38 ± 1.11 |
|        | 2016 | 39.03 ± 1.87 | 114.02 ± 1.06 | 3.97 ± 1.41 |
|        | 2017 | 39.55 ± 1.21 | 109.73 ± 0.69 | 4.11 ± 1.06 |
|        | 2018 | 43.28 ± 1.91 | 112.93 ± 0.74 | 3.73 ± 1.13 |
|        | 2019 | 44.02 ± 1.69 | 111.74 ± 0.97 | 3.51 ± 1.14 |
| Autumn | 2013 | 49.25 ± 2.72 | 228.72 ± 1.46 | 4.23 ± 1.70 |
|        | 2014 | 40.20 ± 1.38 | 234.60 ± 0.93 | 2.32 ± 1.06 |
|        | 2015 | 48.99 ± 1.03 | 224.01 ± 0.87 | 3.17 ± 0.95 |
|        | 2016 | 36.33 ± 1.25 | 232.18 ± 0.77 | 2.21 ± 0.75 |
|        | 2017 | 43.59 ± 1.93 | 227.37 ± 0.96 | 3.28 ± 1.17 |
|        | 2018 | 33.70 ± 2.43 | 240.61 ± 1.30 | 3.89 ± 1.40 |
|        | 2019 | 33.11 ± 1.69 | 237.96 ± 0.79 | 3.03 ± 1.09 |

### 3.2.2. Interannual Variation Trend of Siberian Roe Deer Molting from 2013 to 2019

The results showed that spring SOM and EOM of SRD from 2013 to 2019 in our study area were both advanced. However, in autumn, SOM was delayed, but EOM was advanced (Figure 3).

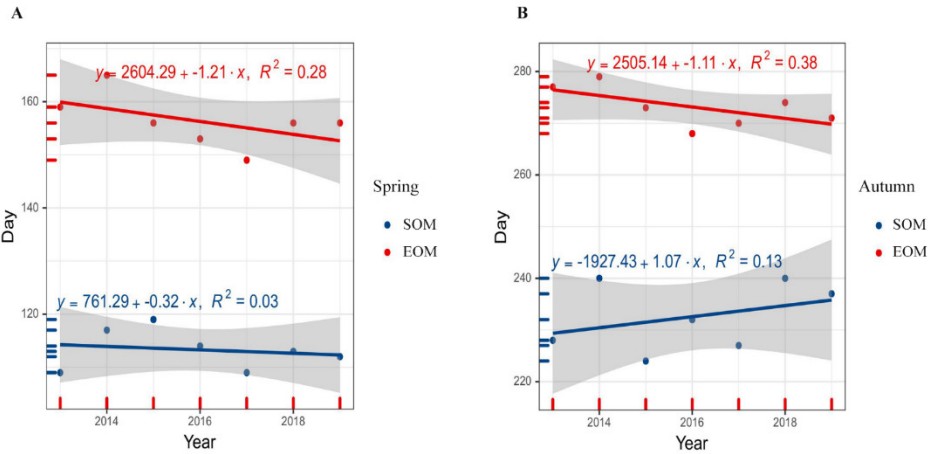

**Figure 3.** Interannual variation of Siberian roe deer spring (**A**) and autumn (**B**) molting 2013–2019. SOM: start date of molting, EOM: end date of molting.

### 3.3. Relationship between Meteorological Factors and Forest Leaf Phenology

According to the correlation analysis results, the length of the optimal period (LP) of meteorological factors affecting FLP was determined (Supplementary Table S4). The results showed that there was a positive correlation between temperature and SOS (r = 0.73). While temperature negatively correlated with EOS (r = −0.65). Precipitation has a great influence on EOS (r = 0.25) and is positively correlated.

The coefficients of the multiple regression equations are all larger than those of the single regression, indicating that the multiple regression has a better fitting effect (Table 3). It can be seen that the relationship between FLP and meteorological factors is not a single linear correlation, but the result of the joint action of a variety of meteorological factors. The results show that temperature is the most important factor in determining FLP, and precipitation also has a certain influence on FLP.

**Table 3.** Regression analysis statistics of forest leaf phenological period and meteorological factors.

| Forest Leaf Phenology | Temperature ($R^2$/$p$ Value) | Sunshine Duration ($R^2$/$p$ Value) | Precipitation ($R^2$/$p$ Value) | Multiple Regression ($R^2$/$p$ Value) |
|---|---|---|---|---|
| Start of growing season | 0.889/<0.05 | 0.083/>0.05 | 0.197/>0.05 | 0.976/<0.05 |
| End of growing season | 0.792/<0.05 | 0.130/>0.05 | 0.807/<0.05 | 0.882/<0.05 |

### 3.4. Relationship between Meteorological Factors and Siberian Roe Deer Molting

After removing the partial correlation of meteorological factors, the correlation analysis was carried out on the meteorological factor, SOM, and EOM of SRD (Table 4). The main meteorological factors affecting spring SOM and EOM of SRD were the mean ssd from winter solstice to SOM (r = −0.739, $p$ = 0.02) and mean ssd from winter solstice to EOM (r = −0.729, $p$ = 0.03). The main meteorological factors affecting autumn SOM and EOM of SRD was the mean ssd from summer solstice to SOM (r = 0.794, $p$ = 0.03) and ssd in October (r = 0.907, $p$ = 0.04), respectively.

**Table 4.** Correlation results of meteorological factors and Siberian roe deer molting.

| Season | Meteorological Factors | Start Date of Molting (r/$p$ Value) | End Date of Molting (r/$p$ Value) |
|---|---|---|---|
| Spring | Mean temperature from winter solstice to start date of molting/end date of molting | 0.206/>0.05 | −0.662/>0.05 |
| | Mean sunshine duration from winter solstice to start date of molting/end date of molting | −0.739/<0.05 | 0.729/<0.05 |
| | Sunshine duration of start date of molting/end date of molting | −0.409/>0.05 | 0.714/>0.05 |
| | Mean temperature of start date of molting/end date of molting | −0.377/>0.05 | −0.559/>0.05 |
| | Mean temperature in March | 0.243/>0.05 | 0.139/>0.05 |
| | Mean temperature in April | −0.203/>0.05 | −0.496/>0.05 |
| | Mean temperature in May | 0.359/>0.05 | −0.201/>0.05 |
| | Sunshine duration in March | −0.291/>0.05 | −0.474/>0.05 |
| | Sunshine duration in April | −0.257/<0.05 | 0.167/>0.05 |
| | Sunshine duration in May | −0.460/<0.05 | 0.578/<0.05 |
| Autumn | Mean temperature from winter solstice to start date of molting/end date of molting | 0.369/>0.05 | 0.615/>0.05 |
| | Mean sunshine duration from summer solstice to start date of molting/end date of molting | −0.794/<0.05 | 0.862/<0.05 |
| | Sunshine duration of start date of molting/end date of molting | 0.665/>0.05 | 0.637/>0.05 |
| | Mean temperature of start date of molting/end date of molting | 0.525/>0.05 | −0.473/>0.05 |
| | Mean temperature in July | 0.105/>0.05 | 0.664/>0.05 |
| | Mean temperature in August | −0.451/<0.05 | −0.526/>0.05 |
| | Mean temperature in September | −0.378/>0.05 | −0.827/>0.05 |
| | Mean temperature in October | −0.493/>0.05 | 0.712/>0.05 |
| | Sunshine duration in July | −0.446/>0.05 | −0.750/>0.05 |
| | Sunshine duration in August | −0.723/<0.05 | −0.342/>0.05 |
| | Sunshine duration in September | 0.721/>0.05 | −0.750/>0.05 |
| | Sunshine duration in October | 0.475/>0.05 | 0.907/<0.05 |

*3.5. Synchronization between Forest Leaf Phenology and Siberian Roe Deer Molting*

The wavelet analysis results showed that there were two obviously synchronic periods (180-day and 365-day) in FLP and SRD molting, which were seasonal periods and entire annual periods respectively. The average power curve for FLP and SRD molting showed two wave crests and the maximum value of average wavelet powers appeared at the temporal scale of 365 days (Figure 4).

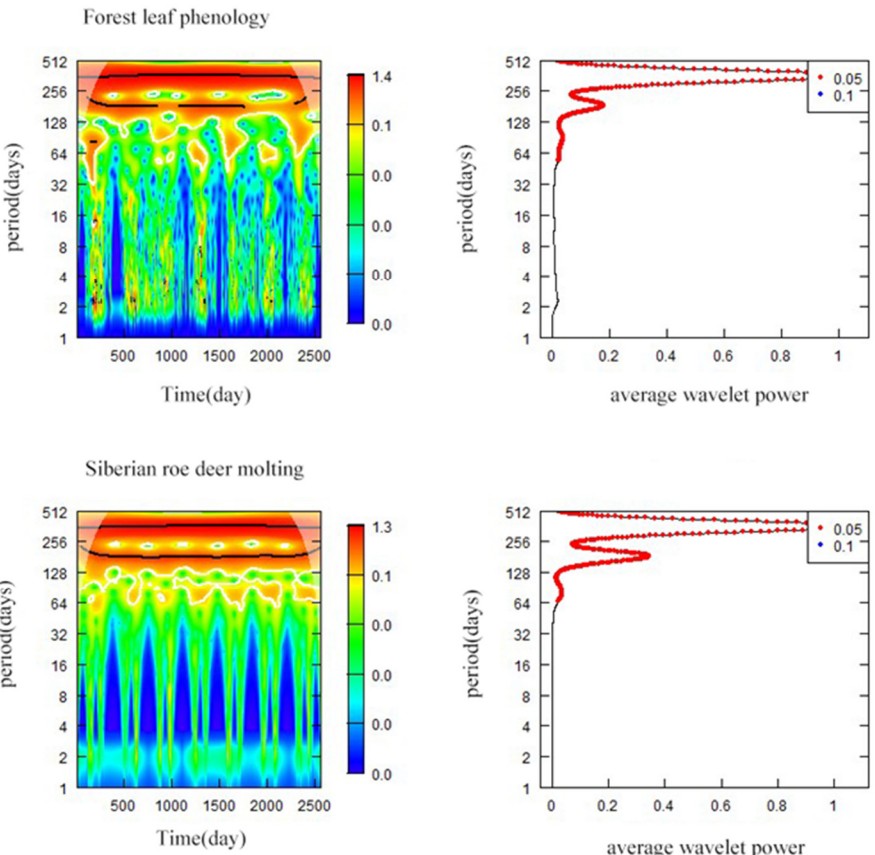

**Figure 4.** Wavelet power spectra and average power based on forest leaf phenology and Siberian roe deer molting 2013–2019. "Period" is the temporal scale (days) that was used for wavelet decomposition, and "index" refers to the days from the start to the end of the research period, which lasted for 2556 days from 1 January 2013 to 31 December 2019.

Cross−wavelet analysis revealed that FLP and SRD molting were highly synchronous at the research temporal scale. The average cross-wavelet power map showed two wave crests and the maximum value of average cross-wavelet powers appeared at a temporal scale of 365 days (Figure 5).

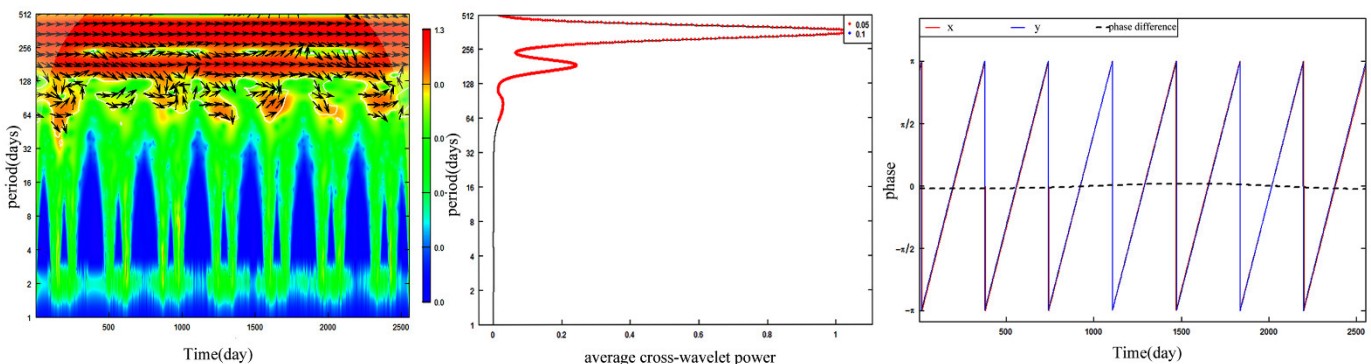

**Figure 5.** Cross−wavelet power spectra, average cross-wavelet power, and phase difference between forest leaf phenology and Siberian roe deer molting. Phase and phase differences were analyzed using a period of 365 days. X represents Siberian roe deer molting ratio and y represents forest leaf phenology.

Seasonal synchrony between FLP and SRD molting showed that the time lag of synchronization fluctuated slightly from year to year in spring and was led by FLP, but their time lag of synchronization in autumn was led by SRD molting (Figure 6).

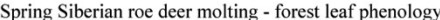

Spring Siberian roe deer molting - forest leaf phenology

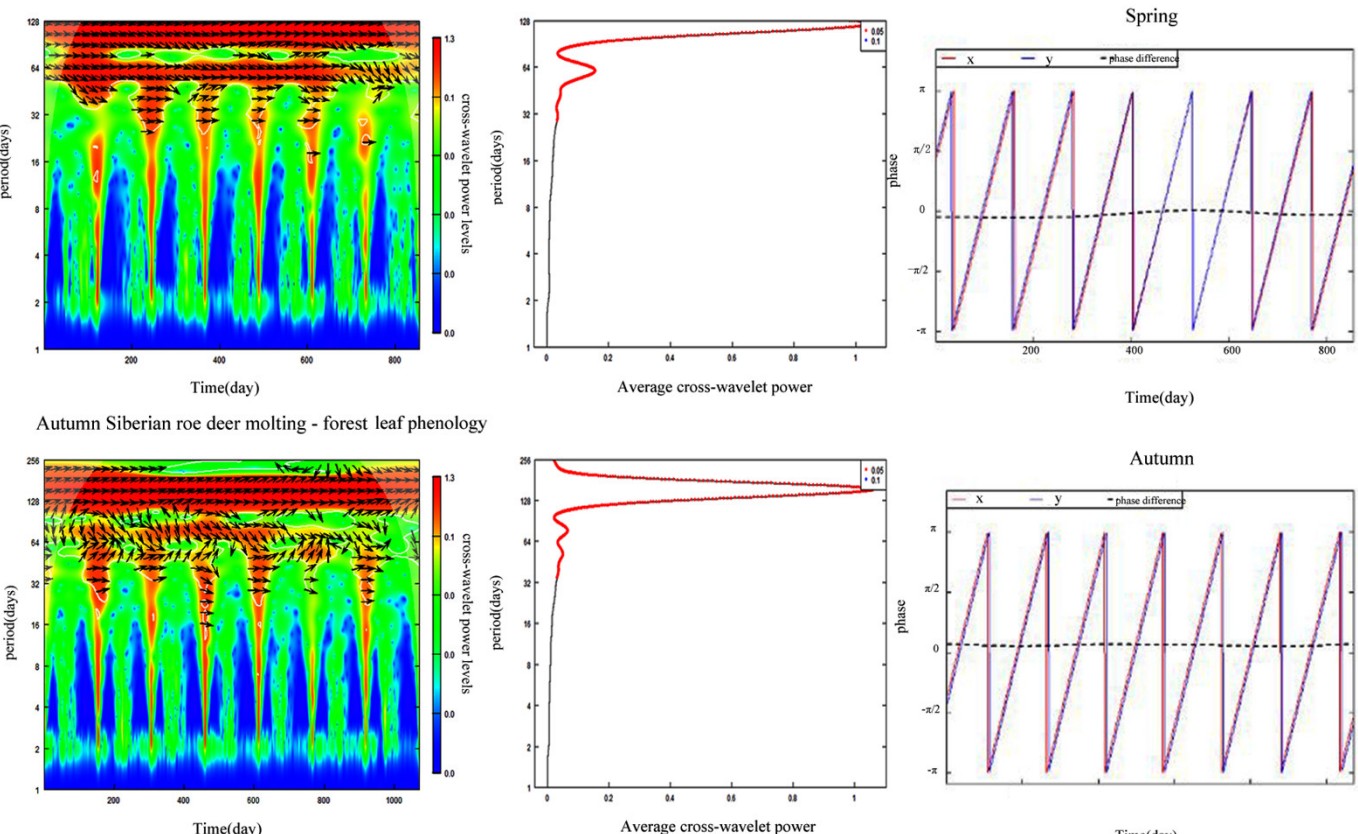

**Figure 6.** Cross−wavelet power spectra, average cross−wavelet power, and phase difference between forest leaf phenology and Siberian roe deer molting. Spring phase and phase differences were analyzed using a period of 122 days, and autumn phase and phase differences were analyzed using a period of 153 days. X represents the Siberian roe deer molting ratio and y represents forest leaf phenology.

We performed pairwise cross-wavelet analysis on FLP, SRD molting, temperature, and sunshine duration (Figure 7), and the results showed that the correlation factors maintained a high synchrony and there was an obvious time lag (phase difference). The difference is that the time lags of SRD molting-MT and FLP-MT were always consistent during the research period, while the time lags of SRD molting-ssd and FLP-ssd showed a trend of gradually shortening. In addition, SRD molting-FLP maintained a high synchrony, with no obvious time lag.

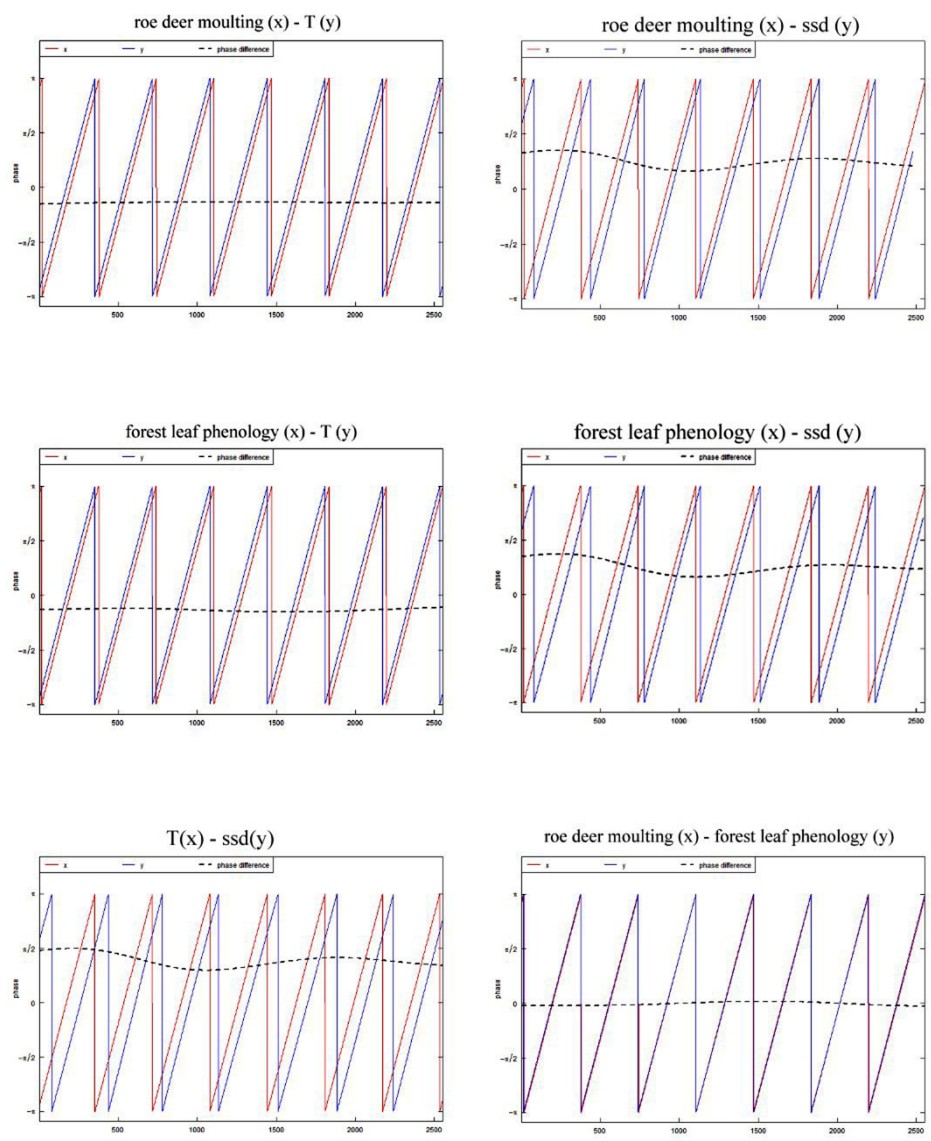

**Figure 7.** Forest leaf phenology and Siberian roe deer molting were analyzed by pairwise cross−wavelet analysis with temperature and sunshine duration.

## 4. Discussion

### 4.1. Effects of Climate Change on Forest Leaf Phenology

Based on camera-trap technology, we extracted the phenophases in the study area from 2013 to 2019. The results of FLP are basically consistent with Li's research on plant phenology in Changbai Mountain using remote sensing technology [27], which confirms the feasibility of using camera-traps data to extract key phenological parameters. As well, the results show that the Klosterman method is more suitable for the data fitting of this study, which is consistent with the results of Zhou [28].

With global climate change, researchers are more and more interested in whether climate change has an impact on phenology such as SOS and EOS [29–31]. The interannual variation trend of FLP results found that SOS of FLP was advanced, EOS was delayed, and LOS was prolonged, which is consistent with the results of most researchers [32–35].

The occurrence of the phenophase is not completely affected by the monthly mean temperature, monthly precipitation, or monthly sunshine duration in a previous month,

but by the mean temperature (precipitation, sunshine duration) in a period before the occurrence of the phenophase [21]. The meteorological factors during this period have the highest correlation with the phenophase [36,21]. Numerous researchers have revealed that the increase in temperature in the early stage of phenology will lead to an earlier phenophase start and a later phenophase end, resulting in a longer growing season [37–39], which is consistent with the conclusion of our results. The mechanism of meteorological factors acting on forest phenology is complex and diverse, not just the result of a single meteorological factor. In the future, we will focus on studying the mechanism and model of the impact of climate change on phenology.

### 4.2. Effects of Climate Change on Siberian Roe Deer Molting

Previous studies have focused on captive animal molting [40–42]. Nowak et al. [17] used camera-traps to quantify and measure the molting sequence of mountain goat (*Oreamnos americanus*), which is the first study in the world on wild animal populations molting, making automatic camera monitoring of wild populations' seasonal molting a reality. Subsequently, some researchers capture the wild animals' seasonal molting by automatic cameras [43,44]. Camera-trap technology has great potential to facilitate the study of mammalian seasonal molting, which not only can save time and cost, but also can measure animals molting at different spatiotemporal scales [41,44]. The results of this study are generally similar to the results of Hua [43] and Zhang [44] on SRD molting in northeast China, indicating that the start date of molting of SRD in spring was in late April and the end date of molting was in early June, and the start date of molting in autumn was in mid-late August and the end date was in late September or early October.

Typical seasonal molting animals are regulated by the photoperiod [45], and some researchers have shown that artificial light control can also make molting advance or delay [46,47]. Our results showed that sunshine duration from the winter solstice to the molting period was the main factor affecting the start or end of SRD molting, which is consistent with other researchers' results [48,49]. In our study, the temperature is also one of the important factors affecting SRD molting, which is contrary to Zhang's results that show that the start and end of SRD molting are independent of temperature [44]. The role of each meteorological factor is different from the contribution of SRD molting, and the influencing mechanism is also very complicated. Existing research is not enough to prove how many meteorological factors affect SRD molting, and future studies will focus on it.

### 4.3. Effects of Climate Change on Phenological Synchrony of Plants and Animals

Animals and forest plants have formed a relatively stable phenological synchronization pattern in the long-term evolution process, and the phenological coordination of animals and plants is an important basis for maintaining the function and stability of communities and ecosystems [2]. Many studies have proved that the occurrence of animal and host plants phenology maintains a high synchronization [50,51]. However, due to global climate change, the differences in the mechanisms and rates of adaptation of animals and plants to climate change may lead to their phenological synchronization pattern being broken, which affects the survival of species and the stability of communities and ecosystems [13,14]. Some studies have found that the seasonal molting of animals is mainly driven by photoperiod [45], while the phenophase of forest vegetation is mainly affected by temperature [52]. Therefore, the changes in the synchrony between temperature and sunshine duration caused by climate change may result in a potential phenology mismatch, thereby threatening the stability of the ecosystem.

The ability to continuously monitor by camera-traps gives us the opportunity to simultaneously understand long-term trends in SRD molting and FLP, and thereby study the profound effects of climate change on their synergy. The analysis presented here suggested that there was substantial synchrony in autumn phenology of SRD molting and FLP, as measured by wavelet analysis. However, the time lag of SRD molting and FLP in spring fluctuates obviously, and the spring synchrony is obviously affected by climate

change. The difference in the order of occurrence of SRD molting and FLP in spring and autumn also implies that although the changes between the two are highly synchronized, the synchronization mechanism in spring and autumn may be different, thus it is necessary to carry out relevant research in different seasons.

Since the main driving factors of SRD molting and FLP are different, we think that, in theory, climate warming will lead to the advance of FLP, and since the SRD molting and the photoperiod do not change much, we expected a mismatch in their synchronization. The results showed that the time lag of synchronization fluctuated slightly from year to year in spring and was led by FLP, but the time lag of synchronization in autumn was stable and led by SRD molting. There may be two potential explanations for these. One is that SRD molting is mainly affected by photoperiod, but to a certain extent, SRD can make certain adaptive changes to environmental changes. the climate warming exceeds the SRD's adaptive capacity, it will inevitably lead to inconsistencies between SRD molting and FLP, thus threatening the survival of SRD. Another explanation could involve a combination of temperature and photoperiod, which can drive SRD molting, but the results cannot be detected due to monitoring technology and time series issues. Nonetheless, it is well established that photoperiod and, to a lesser extent, temperature, control molting phenology [53–56]. Whatever the reason, our results clearly demonstrate the symptoms of climate change impacts on synchronization between SRD molting and FLP. Moreover, future needs to advance this understanding including paying more attention to other mammals' responses to climate change.

## 5. Conclusions

Wavelet analysis indicated that synchrony between Siberian roe deer molting and forest leaf phenology was broken in spring, but in autumn they remain highly synchronized, indicating a potential seasonal phenology mismatch, revealing the influence of climate change on Siberian roe deer molting and forest leaf phenology. We suggested that it is necessary and important to conduct the research in different seasons. Meanwhile, our study provides a new insight and a feasible approach to synchronization between forest leaf phenology and ungulate molting using camera traps data to explore global warming issues.

**Supplementary Materials:** The following supporting information can be downloaded at: www.mdpi.com/article/10.3390/rs14163901/s1. Figure S1: Comparison of Max, Spline and Night method to filter data. The red, green and blue dot represents the result of Night, Spline and Max method, respectively.; Figure S2: Comparison of 4 growth curve fitting methods 2013–2019.; Figure S3: Comparison of 4 growth curve fitting methods and parameter extraction methods 2013–2019; Table S1: Meteorological factors used for analysis; Table S2: Number of images and videos collected of Siberian roe deer; Table S3: Data of Siberian roe deer molting events in study area; Table S4: length of optimal period (LP) of forest leaf phenology (FLP).

**Author Contributions:** Conceptualization, H.C. and Y.H.; methodology, H.C. and Y.H.; software, H.C and X.L.; formal analysis, H.C.; investigation, H.C., D.W. and Z.L.; data curation, D.W., Z.L. and J.Q.; writing—original draft preparation, H.C. and Y.H.; writing—review and editing, Z.L., J.Q., N.J.R., H.S. and G.J.; supervision, H.S. and G.J.; project administration, H.S. and G.J.; funding acquisition, Y.H., H.S. and G.J. All authors have read and agreed to the published version of the manuscript.

**Funding:** This research was funded by National Natural Science Foundation of China, grant number 32060307 and National Natural Science Foundation Surface Project of China, grant number 31872241.

**Data Availability Statement:** Not applicable.

**Acknowledgments:** We thank Hong Wang, Xue Liang, Biyu He, Xiaoxia Yuan and Xue Zhang for their care and help.

**Conflicts of Interest:** The authors declare no conflict of interest.

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
