# Peer review of "Wavelet Analysis Reveals Phenology Mismatch between Leaf Phenology of Temperate Forest Plants and the Siberian Roe Deer Molting under Global Warming"

_remotesensing, doi:10.3390/rs14163901_

Round 1
Reviewer 1 Report
The manuscript aims to address a possible mismatch between plant phenology and molting of Siberian roe deer as a consequence of global warming using camera traps. While I find the topic of the paper interesting, I think the authors can improve their manuscript. First of all, I strongly suggest the review of a native English speaker to address several language issues.
Below there are some general comments and attached there is a pdf with specific comments (comments visible when opening the file with Adobe).
Introduction
I believe the authors should review their manuscript keeping in mind the aims and scope of the journal. The main topic in the introduction should be the application of remote sensing techniques (in this case camera traps) to environmental science (phenology mismatch) highlighting how this approach can benefit the scientific community (see my general comment for the discussion). Once addressed this more general topic, I would reorganize the current information in the introduction trying to go from general to specific, e.g., global warming and climate change, interaction/synchronization between mammal species and plants in terms of leaf production, food and camouflage, how climate change affects this interaction in other species, the specific case of the SRD and why you think it is a good model species or how in its environment the possible mismatch can affect the community they belong to. The information is mostly there, but it is scattered and not well organized.
Materials and Methods
The information provided is not enough for the reader to replicate your study. See some of the specific comments. Moreover, a paragraph where you tell the reader more about the study area would be helpful. Where is it? Is it a forest? What type of habitat? Main tree species? Climate? etc etc...
Results
The way you present results should follow the order of topics you used to present materials and methods. Moreover, since we talk about climate, I expected to see at least in SM a table with average values of temperature (maybe also T min and max) and precipitation per year per season.
Table and Figure captions
They need to be complete. I should be able to look at the table or figure and reading the caption understand what it means.
Discussion
In the discussion 4.2 you discuss if camera traps have been used before to address your same question and what the benefits are of using this approach. What is the final goal of your study and how can other people apply it in other circumstances? You should expand a little more to better fir the journal’s aims and scope.
Line 284-296 this paragraph should all be in the introduction.
Conclusions
They are short and a repetition of what you just said in the previous paragraph. I would remove or expand.

Author Response
Dear reviewer,
Thank you very much for your valuable advice. I have revised your specific comments point by point, and the following is my reply to your Pointuestion. In addition, the revised draft is attached. Please check them. Thank you.
Point 1: First of all, I strongly suggest the review of a native English speaker to address several language issues.
Response 1: The author Nathan James Roberts is a native English speaker and he has checked my English grammar.
Point 2. Introduction
Response 2: I have reorganized and modified this part in the revised draft according to your opinions. Please see the attachment.
Point 3: Materials and Methods
Response 3: In this section, I added the content, specifying the methods and steps. Please see the attachment.
Point 4: Results
Response 4: I have added a supplementary material about meteorological data (monthly mean temperature, T max, T min, precipitation and sunshine duration). Please see the attachment.
Point 5: Table and Figure captions
Response 5: In this section, I have completed the captions of table and figure. Please see the attachment.
Point 6: Discussion
Response 6: Camera traps have previously been used to monitor plant phenology and animal molt, but they have been done separately. I use an infrared camera to monitor the phenology of plants and animals simultaneously, which can save time and cost, and this is a highlight of my research.
Point 7: Conclusions
Response 7: I have expanded on the conclusions. Please see the attachment.
Thank you again for your pertinent advice. Please check the revised manuscript.

Reviewer 2 Report
I really enjoyed reading this paper. I work on a mammal with seasonal coat colouration change and I had not considered the impact of climate change on this trait, and this paper, which investigates forest leaf phenology as a factor related to climate change that can be linked to coat moulting, has inspired me in my own work.
A first note is that although the English is quite good, there are many small errors throughout - especially misuse of definite articles (e.g. a lot of missing "the" or "a" or "an") and some issues with subject verb agreement- this needs to be checked by an English speaker.
Line 43 - spell out roe deer the first time mentioned; also the term "choose" is used here - have coevolved to? have been selected to?
Line 64 - I appreciate this solid research question - as the introduction is relatively short, are there any previous studies that showed a mismatch that led to this prediction?
Line 80 - Figure1 - ensure that the figure heading stands alone from the text including the name of the study region and period that the work took place (please consider this for other tables and figures too - instead of "study area for example state WHERE)
Line 83: SRDs' coats (plural possessive)
Line 84 - this is an interesting method for the coat score - I noticed that the package is for moulting in birds - has it been used for mammals before as the moulting pattern is quite different - if so, please include a citation - if not please explain the process of generalising the method to work with mammals, as this will help others
Section 2.7 - I am not familiar with this analysis but it is very interesting and I would like to try it!
Line 169 - did you make an effort to count different individuals? or did you exclude individuals on the same camera on the same day?
The figures of the wavelets are different sizes and different orientations making them hard to compare - can they all be in a single panel? Or certainly be scaled to the same size and direction?
Line 241 - before going into the specifics, give an overview of the results - you start to do this and then skip to very specific areas.
Line 263 - this is the section I felt was missing from the introduction - I would suggest to edit and move to the intro leaving the results aspect of these studies to discuss the comparison
Line 309 - change to. - leads to synchronisation being destroyed - this is a really important part of this study and is very interesting - I feel this needs to be the headline in the first paragraph of the discussion and then come back to it here.
Conclusions (check spelling) - is this part of the journal style? as this is the statement that could headline the discussion. I find the recommendation for the studies to be completed in different seasons also apt and important.
Author Response
Dear reviewer,
Thank you very much for your valuable advice. I have revised your specific comments point by point, and the following is my reply to your question. In addition, the revised draft is attached. Please check it. Thank you.
Point 1: Language issues.
Response 1: The author Nathan James Roberts is a native English speaker and he has checked my English grammar.
Point 2: specific comments
Response 2: I have accepted and revised your comments one by one. You can check them in my revised draft. Thank you.
Point 3: Introduction
Response 3: I have reorganized and modified this part in the revised draft according to your opinions. Please see the attachment.
Point 4: Figure and table
Response 4: I have revised all the pictures and tables according to your comments and made corresponding supplements. Please see the attachment.
Point 5: methods
Response 5: Measurement and determination of Siberian roe deer molting, I combine Nowark et al [17] and UZ model [19,20] because of limited molting events. An appropriate model can be selected according to the molting type. In this study, our samples included the molting ratio of roe deer before molting, during molting and after molting, so type2 was selected as molting model to ensure the highest accuracy of the results.
Point 6: did you make an effort to count different individuals? or did you exclude individuals on the same camera on the same day?
Response 6: Yes, I count different individuals and exclude individuals on the same camera on the same day. In the SM table S2 and table S3, Effective molting events mean the molting of different individuals on the same day.
Thank you again for your pertinent advice. Please check the revised manuscript.

Reviewer 3 Report
I did some minor comments/corrections in the ms. Please, avoid the use of acronyms to improve the fluency in the read of the ms.

Author Response
Dear reviewer,
Thank you very much for your valuable advice. I have revised your specific comments point by point, and the following is my reply to your question. In addition, the revised draft is attached. Please check it. Thank you.
Point 1: Language issues.
Response 1: The author Nathan James Roberts is a native English speaker and he has checked my English grammar.
Point 2: specific comments
Response 2: I have accepted and revised your comments one by one. You can check them in my revised draft. Thank you.
Point 3: avoid the use of acronyms to improve the fluency in the read of the ms.
Response 3: I have deleted the acronyms. Please see the attachment.
Thank you again for your pertinent advice. Please check the revised manuscript.

Round 2
Reviewer 1 Report
The authors have addressed most of my comments. Still some minor revisions, especially for the English language.
Line 19: processes
In the abstract the location of the study is missing (northeast China)
Line 20: around the world are needed to reveal it
Line 21: we report how… respond
Line 22: and investigate whether it will result in a potential phenology mismatch. (no question mark)
Line 30: remove coma
Line 37-50: move this paragraph to line 89 and integrate it with paragraph 89-93
Line 51: remove and
Line 65: remove even
Line 99: selected a total of
Line 101: …surveys. Camera traps were placed near…..
Line 105: the study area
Line 115: we used
Line 140: I think “i” should be a subscript
Results 3.3 and 3.4: shouldn’t the spearman correlation results also have a p value?
262: incomplete sentence “indicating that their seasonal period and entire annual period”.. what?
Line 264 and 275: remove was
Line 307: remove and
Line 334-336: maybe expand a little more
Line 343: that show that the start
Line 345-346: rephrase in a more appropriate language
Line 364: suggested that there was substantial synchrony in autumn
Line 365: “fluctuates obviously” rephrase
Line 367: order of occurrence of
Line 372-373: and since the SRD molting….., we expect a mismatch in their synchronization
Line 393: “to complete” rephrase
Author Response
Dear reviewer,
Thank you very much for your valuable advice. I have revised your specific comments point by point, and the following is my reply to your question. In addition, the revised draft is attached. Please check it. Thank you.
Point 1: specific comments
Response 1: I have accepted and revised your comments one by one. You can check them in my revised draft. Thank you.
Point 2: about table 3.3 and 3.4
response 2: I add the P value in the table, please check. Thanks for your comments.
Thank you again for your pertinent advice. Please check the revised manuscript.

This manuscript is a resubmission of an earlier submission. The following is a list of the peer review reports and author responses from that submission.